# Conditional prediction of consecutive tumor evolution using cancer progression models: What genotype comes next?

**Juan Diaz-Colunga**[1,2,3], **Ramon Diaz-Uriarte**[1,2]*

**1** Department of Biochemistry, School of Medicine, Universidad Autónoma de Madrid, Madrid, Spain,
**2** Instituto de Investigaciones Biomédicas 'Alberto Sols' (UAM-CSIC), Madrid, Spain, **3** Department of Ecology & Evolutionary Biology and Microbial Sciences Institute, Yale University, New Haven, Connecticut, United States of America

* ramon.diaz@iib.uam.es

**Data Availability Statement:** Data and code for the analyses in this article are available at https:// github.com/rdiaz02/what_genotype_next.

## Abstract

Accurate prediction of tumor progression is key for adaptive therapy and precision medicine. Cancer progression models (CPMs) can be used to infer dependencies in mutation accumulation from cross-sectional data and provide predictions of tumor progression paths. However, their performance when predicting complete evolutionary trajectories is limited by violations of assumptions and the size of available data sets. Instead of predicting full tumor progression paths, here we focus on short-term predictions, more relevant for diagnostic and therapeutic purposes. We examine whether five distinct CPMs can be used to answer the question "Given that a genotype with $n$ mutations has been observed, what genotype with $n + 1$ mutations is next in the path of tumor progression?" or, shortly, "What genotype comes next?". Using simulated data we find that under specific combinations of genotype and fitness landscape characteristics CPMs can provide predictions of short-term evolution that closely match the true probabilities, and that some genotype characteristics can be much more relevant than global features. Application of these methods to 25 cancer data sets shows that their use is hampered by a lack of information needed to make principled decisions about method choice. Fruitful use of these methods for short-term predictions requires adapting method's use to local genotype characteristics and obtaining reliable indicators of performance; it will also be necessary to clarify the interpretation of the method's results when key assumptions do not hold.

## Author summary

Predicting cancer progression would allow for the systematic application of targeted, personalized therapies. In recent years, several methods have been developed to predict cancer evolution from cross-sectional sequencing data, which are increasingly available. However, the quality of these predictions is hampered by violations of the methods' assumptions, often in conflict with the evolutionary dynamics of real tumors. Since predicting the short-term progression of a tumor after its detection could be more relevant

**Funding:** Supported by grant BFU2015-67302-R (MINECO/FEDER, EU) funded by MCIN/AEI/10.13039/501100011033 and by ERDF A way of making Europe and by grant PID2019-111256RB-I00 funded by MCIN/AEI/10.13039/501100011033 to RDU. JDC supported by PEJD-2018-POST/BMD-8960 from Comunidad de Madrid to RDU. The funders had no role in study design, data collection and analysis, decision to publish, or preparation of the manuscript.

**Competing interests:** The authors have declared that no competing interests exist.

from a clinical perspective, we examine the feasibility of short-term predictions, hypothesizing that these could be successful even when long-term predictions are not possible: even if the methods' assumptions are not satisfied in general, they could still hold for specific evolutionary steps. We examined whether 13 methods could accurately predict the short-term evolution of over 25 million simulated tumors, and identified the conditions for predictions to be accurate. We analyzed 25 real cancer data sets and found indications that forecasting the evolution of a tumor could be possible when specific mutants are found in it. Our analysis highlights the importance of conditioning predictions on the detected tumor composition, and opens new avenues for developing and adapting methods aimed to predict cancer evolution.

## Introduction

Predicting the evolution of a tumor is a critical goal of cancer biology. In general, clinical decisions are guided by cytology and molecular markers used for cancer staging, but these can be unreliable due to sampling problems and the inherent heterogeneity of tumors [1]. The emergence of next-generation sequencing tools has brought promise for a better understanding of the dynamics of cancer evolution [2], and has fueled the development of methods aimed to predict tumor progression [3–11]. Unfortunately, the stochastic components in key factors that govern cancer evolution (mutation, genetic drift, or clonal selection [12–14], as well as non-genetic variability [15] and the tumor microenvironment [16]) fundamentally limit predictability –this is also common to other problems [17, 18] such as the evolution of antibiotic resistance [19] or virulence [20]. Nonetheless, repeated tumor progression across patients indicates that prediction must be possible to some extent [21]. Even small increases in predictive power could be of critical importance for the design of treatment strategies [22].

Cross-sectional high-throughput data are widely available for many cancer types, and Cancer Progression Models (CPMs) [23, 24], originally developed to identify restrictions in the order of accumulation of mutations in tumors using cross-sectional data, could thus be key tools for predicting cancer progression. CPMs (implicitly) encode all the possible mutational trajectories that are compatible with the restrictions and, for some methods, provide probabilities of different paths (Fig 1A and 1B). However their performance to infer the complete evolutionary trajectories, from the root genotype with no driver mutations to a final, fixated genotype, is limited [25, 26]; this has been attributed mainly to violations of the underlying assumptions of the models (fitness landscapes with a single fitness maximum in the genotype with all loci mutated, strong selection week mutation regime [27]) and the difficulty to acquire large, unbiased data sets to power them [25].

Predicting the complete evolutionary path, however, might not be the most relevant objective. Detection of a specific tumor composition at a given point in time automatically discards all mutational pathways that do not visit the observed state, reducing the amount of possible evolutionary trajectories that can be followed from then on and thus potentially facilitating the task of predicting further tumor progression. In this work we address the question of how accurately CPMs can predict short-term cancer evolution after a genotype is detected: what is the state that comes consecutively after or "what genotype comes next". Even if the predictability of the complete path is low because of violations of method assumptions, genotype-specific features could lead to local improvements in short-term predictability [28–30]. Moreover, in contrast to the more general, non-local descriptors, these genotype-specific factors can provide more detailed insight regarding the requirements for the methods to be reliable.

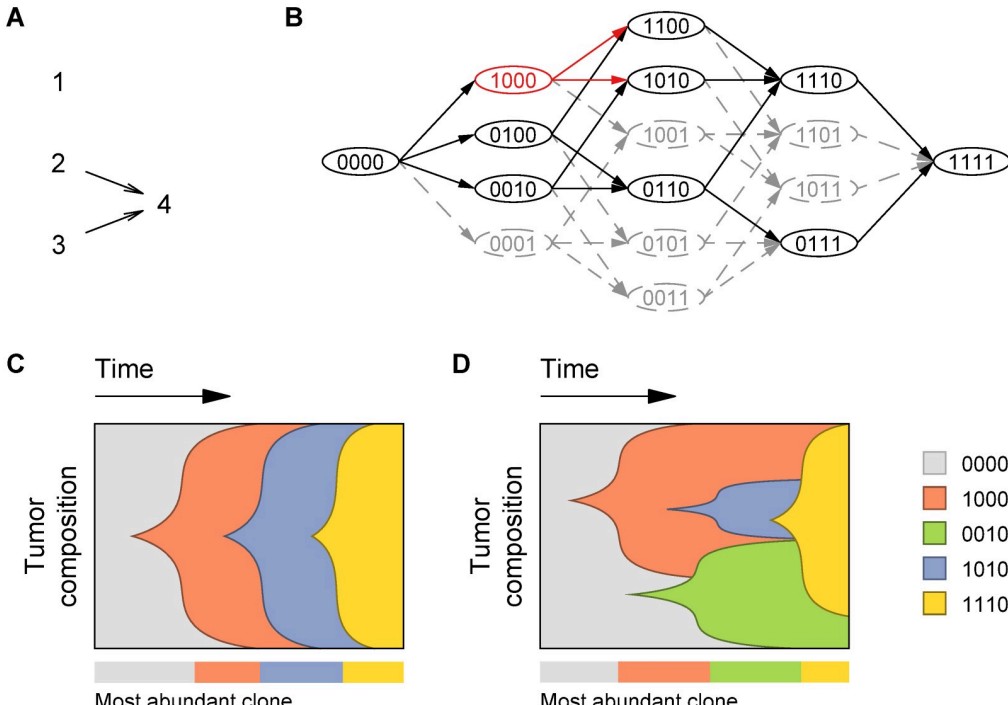

**Fig 1. Inferring evolutionary trajectories using cancer progression models.** A: Example of a directed acyclic graph (DAG) of restrictions in the order of accumulation of mutations, analogous to the ones produced by most CPMs, such as CBN, MCCBN, OT, CAPRESE, and CAPRI. Given four driver genes (numbered 1 to 4), the DAG indicates the constraints in the order of accumulation of mutations (arrows). Here, a mutation in the 4th loci is conditioned to the 2nd and 3rd ones having been previously mutated. MHN uses a network to encode both promoting and inhibiting effects between pairs of events; the network could be identical to the one in A (meaning that mutations 2 and 3 have a promoting effect on mutation 4), but it could also include inhibiting effects; in contrast to DAGs of restrictions, MHNs do not denote deterministic dependencies. B: Restrictions in the order of accumulation of mutations limit the amount of available evolutionary trajectories and stochastic dependencies change the probabilities of trajectories. In this network, nodes represent different genotypes of four driver genes and edges connect those that are a single mutation away. Genotype names are assigned according to which genes are mutated (1) or not mutated (0). Without restrictions many mutational pathways are possible (any route through both solid and/or dashed edges, from left to right). However, only a subset of genotypes satisfy the constraints in panel A (solid nodes); thus, only trajectories connecting these are predicted to be accessible pathways (solid edges). For example, given a specific genotype (1000, red node), restrictions in the mutational order limit the possible clones that can evolve from it consecutively after (red edges). With CBN, MCCBN, and OT, in addition to the set of accessible pathways, it is possible to obtain transition probabilities along the black lines. Under the MHN model, as there are no deterministic dependencies, all transitions (to genotypes with one additional mutation) are possible, but transitions affected by inhibitory stochastic dependencies will have very low probability (in this simplified depiction they would be the gray lines) and those affected by promoting stochastic dependencies will have higher probability (black lines); MHN provides transition probabilities for all the lines in the figure. C: Example of cancer progression with four driver genes under SSWM conditions. Progression towards the fitness maximum occurs through a defined sequence of parent-child genotypes, each one of them establishing as the most abundant rapidly after evolving. D: Example of cancer progression with four driver genes under conditions of no SSWM. Here, the sequence of parent-child genotypes from the wild-type with no driver mutations to the final one (i.e. the Line Of Descent or LOD) is: 0000 → 1000 → 1010 → 1110. Notice that in the most general case, the ultimately fixated genotype need not be the one with all driver mutations (1111). Some genotypes can be most abundant in the tumor during prolonged periods of time despite not being part of the LOD (0010, in green), and genotypes that do belong to the LOD may never be most abundant (1010, in blue). In all models and figures we do assume that there are no back mutations and that crossing valleys in the fitness landscape using a single multi-mutation step is not possible.

In fact, conditional predictions on the observed genotype are arguably much more relevant than complete evolutionary path predictions for both SMART (sequential, multiple assignment, randomized trial) designs in dynamic treatment regimes for personalized medicine [31, 32], and for adaptive therapy (e.g. [33–35]). Most cancer treatments aim for an immediate

response that maximizes the killing of diseased cells. However, it has been pointed that this approach may favor the proliferation of resistant clones by removing competition, raising the need for alternative, more targeted therapeutic strategies [34, 36, 37]. For example, intermittent androgen deprivation cycles have been suggested to prevent or delay the expansion of resistant mutants in prostate cancer [38]. Furthermore, multiple studies have aimed to "steer" the evolution of tumors towards states of increased treatment susceptibility, for instance through the sequential application of immunotherapy and chemotherapy in small cell lung cancer [39]. Even when this "steering" is not possible, predicting the short-term evolution of a tumor could allow physicians to anticipate treatments aimed to suppress the proliferation of specific, more aggressive clones predicted to evolve consecutively [35]. Short-term predictions could also aid in patient stratification for personalized medicine, e.g., a clinical study has suggested that forecasting subsequent mutations in precursor gliomas could serve to classify tumors into later-developing glioblastoma subgroups [40].

The question of what genotype comes next, or what is the state that comes consecutively after, requires that we carefully define the meaning of "consecutive" states in this context. If cancer progresses under conditions of strong selection and weak mutation (SSWM) [27], only evolutionary trajectories of increasing fitness are possible, and the evolution of a new, higher-fitness clone results in rapid expansion and effective exclusion of previous ones before any new mutations take place. In this situation, there is a well defined sequence of genotypes with progressively increasing fitness in the path to tumor fixation, with the most abundant clone at every time being a direct descendant of a previously most abundant one itself (Fig 1C). But, in cancer, large population sizes and high mutation rates can prevent successful clonal sweeps [41, 42] so that instead of SSWM, clonal interference and stochastic tunneling [43–47] become common: multiple clones can coexist at significant fractions for prolonged periods of time, and new higher-fitness clones can evolve from low fitness parents that represented a minority of the population (Fig 1D).

To formulate the question of "what genotype comes next" so that it does not depend on the SSWM assumption we designate an *observed* genotype as that which is most abundant in the tumor at the time of sampling. On the other hand, a Line of Descent (LOD) is defined as a sequence of parent-child genotypes from the wild-type to the (possibly local) fitness maximum where fixation occurs [44]. By construction, assuming that mutations accumulate one by one and cannot be lost, each genotype in the LOD has all the mutations of the previous one plus exactly one more (as it directly descends from it). The LOD has no more than one genotype with a given number of mutations, whereas in general two or more observable genotypes could share the same mutation count (e.g. in Fig 1D both the red and green genotypes have one mutated driver and are observable). If the evolutionary process does not conform to SSWM, not all genotypes in the LOD are necessarily observable. From a clinical perspective, it is relevant to use the information that is detectable to unveil the underlying dynamics of cancer evolution, encoded in the LOD. In other words, we want to make a prediction regarding *what comes next in the LOD* based on the *observation* of a genotype. We thus formulate the following general question: *Given that a genotype with* n *mutations has been observed, what genotype with* n + 1 *mutations is in the LOD?* Formulated like this, the question is unambiguous and determined. Under SSWM conditions it can be expressed simply as: *given that a certain genotype has been observed, what genotype comes next in the path to tumor fixation?* However, it is important to emphasize that even if there is no SSWM the more general question remains well defined.

To address this question, we use a collection of evolutionary simulations from which we obtain the true paths of tumor progression. For every simulated process we keep track of the most frequent genotypes. Then for each one of them we ask what genotype has exactly one

more mutation while also being a direct ancestor of the ultimately fixated clone. Averaging over all simulations we obtain a set of frequencies which represent the conditional probabilities that we are interested in. We then examine five CPMs: Mutual Hazard Networks (MHN) [3], Conjunctive Bayesian Networks (CBN) [4–6], Oncogenetic Trees (OT) [7, 8], CAncer PRogression Inference (CAPRI) [9, 10] and CAncer PRogression Extraction with Single Edges (CAPRESE) [11], some of which have more than one variant (giving a total of 13 different methods). We provide these methods with cross-sectional samples taken from the evolutionary simulations and compare their predictions with the true probabilities. We explore the factors that affect performance, focusing on those factors that depend on the properties of the observed genotype. We want to understand whether CPMs can be used to predict short-term evolution; for this question, the key response variable is the best possible performance over all CPMs considered. Although we do address method choice as a secondary question, this is not our main objective *per se*. We discuss the consequences of our results for the analysis of cancer data by analyzing 25 cancer data sets.

## Methods

### Overview

To assess if we can use CPMs for short-term predictions of evolutionary processes, we have compared predicted and true answers to the question *Given that a genotype with* n *mutations has been observed, what genotype with* n *mutations is in the LOD?* under an extensive set of simulated scenarios. First, we simulated a large number of tumor evolution trajectories under scenarios with varying departures from the assumptions of CPM methods; next, we obtained a large number of simulated tumor evolution trajectories. Then, we emulated cross-sectional sampling of the simulations, with varying sampling schemes and sample sizes. These steps are described in the section "Simulated evolutionary processes and sampling" of the Methods. The transition probabilities that provide the true answer to the question *Given that a genotype with* n *mutations has been observed, what genotype with* n + 1 *mutations is in the LOD?* are explained in section "Transition probabilities from evolutionary simulations" of the Methods. To obtain the CPM predictions we analyzed each of the sampled data sets with a total of 13 different CPM variants and obtained the predictions as explained in section "Transition probabilities from CPMs" of the Methods. Next, we measured how close the predictions were to the truth (Methods section "Quantification of similarity between predicted and true transition proabilities") and, finally, we analyzed the major factors that affect the similarity between predictions and truth (Methods section "Factors that affect predictive performance: linear mixed-effects model trees"). To examine the implications of our results for the analyses of cancer data, we have analyzed 25 cancer data sets by comparing the predictions of different methods and matching them with the patterns seen in the simulated data, as described in section "Cancer data sets" of the Methods.

### Simulated evolutionary processes and sampling

Simulated data were taken from [25], where complete details are provided. In summary, simulations were conducted under evolutionary scenarios that differed in the number of genes, the type of fitness landscape, the initial population size, and the mutation rates. Landscapes were of either 7 or 10 genes. Fitness landscapes were of three types: representable, local maxima and Rough Mount Fuji (RMF). The three types of fitness landscapes incorporate increasing departures from the assumptions of most CPMs: for the representable fitness landscapes a DAG of restrictions exists with the same accessible genotypes and accessible mutational paths; for local maxima fitness landscapes the set of accessible genotypes can be represented by a DAG of

restrictions, but there are local fitness maxima and the fitness graph has missing paths [25]; the genotype with all genes mutated might or might not be the genotype with largest fitness. The RMF fitness landscapes [43, 48, 49] usually have multiple local fitness maxima and considerable reciprocal sign epistasis so not even the set of accessible genotypes can be represented by a DAG of restrictions (further details about the fitness landscapes and their correspondence to the methods used are provided in section 1.7 of S1 Appendix). On the above fitness landscapes evolutionary simulations were run. Three different populations sizes ($2 \times 10^3$, $5 \times 10^4$ or $1 \times 10^6$) and two mutation rate regimes (mutation rate fixed at $10^{-5}$ or uniformly distributed in the log scale between $0.2 \times 10^{-5}$ and $5 \times 10^{-5}$) were used to generate variability with respect to SSWM. For each combination of the above factors, 35 independent replicates were obtained for a total of 1260 landscapes (35 replicates × 2 number of genes × 3 types of fitness landscapes × 3 initial population sizes × 2 mutation rate regimes). Then, for each one of them, 20000 independent evolutionary processes were simulated until fixation of one of the genotypes at a fitness maximum (local or global). The complete history of the simulations was stored so that the true LOD for each simulation was known. Each set of simulations was then sampled under three detection regimes: with equal probability with respect to the size distribution (*uniform* sampling), sampling predominantly early in the tumor progression process (i.e., enriched in *small* tumors), sampling predominantly late in the tumor progression process (i.e., enriched in *large* tumors). When *sampling*, we identified the most abundant genotype and its mutated loci at the chosen time, so the sampling process is consistent with how we defined *observable* genotypes earlier. No sampling error is considered. The full sample consists of three binary matrices of observations (one per detection regime), each with 20000 rows (one observation per simulation) and as many columns as driver genes. Each element of the matrix is either 1 or 0 (true or false) depending on which genes were mutated in which observations. For each detection regime, the complete sample was used to generate five smaller, non-overlapping splits of size 50, 200 or 4000. This produced a total of 56700 data sets (1260 fitness landscapes × 3 detection regimes × 3 sample sizes × 5 splits).

## Transition probabilities from evolutionary simulations

Let us consider a matrix **P** where rows correspond to observable genotypes and columns to genotypes of the LOD. The element $(i, j)$ of the matrix, denoted as $p_{ij}$, represents the conditional probability that the genotype $j$ (with $n + 1$ mutations) is an element of the LOD given the genotype $i$ (with $n$ mutations) has been observed. In the absence of back mutations and when mutations accumulate one by one, $p_{ij}$ is defined only if $i$ and $j$ are exactly one mutation away from each other. Note that the genotype $j$ (element of the LOD and with $n+ 1$ mutations) need not be a direct descendant of $i$ if SSWM does not hold. We built a transition matrix **P** for every fitness landscape that we generated by examining each of the 20000 evolutionary processes that were simulated in the corresponding landscape, listing all observable genotypes (i.e. those that represented the majority of the population at some point during the simulated evolutionary process) and extracting the LOD of every process tracking back the ancestors of the fixated genotype (definition and details in [44]; using implementation in [50]). Each element $p_{ij}$ can thus be interpreted as the fraction of the times that $j$ was in the LOD given that an observation of $i$ had been made. See further details in section 1.5 of S1 Appendix. Additionally, we allow for a possible answer to the question of "what genotype comes next?" to be "none does". The probability of there not being a "next genotype" is associated with the observation of the final, fixated genotype or, alternatively, with an observed transient genotype that has the same or higher mutational load than the final genotype (see further details in section 1.1 of S1 Appendix).

## Transition probabilities from CPMs

General overviews of all the methods used here, except for MHN, are available in [23–25, 51] and detailed descriptions are given in the original references for each procedure (MHN [3], CBN [4, 5], MCCBN [6], OT [7, 8], CAPRI [9, 10], CAPRESE [11]). Briefly, CBN, MCCBN, OT, CAPRESE, and CAPRI try to identify restrictions in the order of accumulation of mutations from cross-sectional data. CAPRESE and OT code these restrictions using trees (i.e., under these two methods each event, such as a mutation, can immediately depend on at most one other event); although the output of OT and CAPRESE is often similar, the procedures are different (in OT weights along edges can be directly interpreted as probabilities of transition [7], whereas CAPRESE uses a probability raising notion of causation based on Suppes' probabilistic causation). CBN, MCCBN, and CAPRI code these restrictions using DAGs (i.e., each event can depend on multiple other events). CBN and MCCBN are very closely related methods: the CBN version we use is H-CBN as described in [4, 5], and MCCBN is an implementation of the CBN model that uses a Monte Carlo Expectation-Maximization (EM) algorithm for fitting the CBN model [6] (instead of the simulated annealing nested within EM of H-CBN); in contrast, CAPRI tries to identify probability raising in the framework of Suppes' probabilistic causation. Common to most of these methods is a model of deterministic dependencies [3], a model of accumulation of mutations where an event (a mutation) can only occur if all its dependencies are satisfied (though different methods allow for small error deviations from this requirement); CAPRI and CAPRESE try to recover "probability raising" relations and, strictly, their trees/DAGs do not encode deterministic dependencies but will be treated as doing so in this paper, since obtaining probabilities of transition from them is not possible otherwise—see details in section 1.2 of S1 Appendix.

For all of these methods, since a DAG of restrictions determines a fitness graph (see Fig 1A and 1B and [25, 51]), we can obtain the set of possible descendants for a genotype directly from this fitness graph. For CAPRESE and CAPRI all transitions were set as equiprobable (as probabilities of transition are not available) but for CBN, MCCBN, and OT we can compute the probabilities of transition between genotypes. CBN and MCCBN return the parameters of the waiting time to occurrence of each mutation given its restrictions are satisfied which allows us to compute transition probabilities between genotypes using competing exponentials (see [6, 25, 26]) and a similar procedure can be used with OT by using the edge weights (though, strictly, the OT model, being an untimed oncogenetic tree does not return these probabilities—see [25]). Mutual Hazard Networks (MHN) [3] differ from the previous methods in that events are modeled by a spontaneous rate of fixation and a multiplicative effect each of these events can have on other events (i.e., pairwise interactions), which allows it to model both enabling and inhibiting dependencies. The output of MHN is a transition rate matrix; as for CBN and MCCBN, we can obtain the probability of transition to each descendant genotype, given a transition, using competing exponentials from the transition rate matrix (see equation 2 and Figure 2 in [3]). Details on software and computation of transition probabilities are provided in sections 1.2 and 1.5 of S1 Appendix.

None of the CPM models considered here allow us to accommodate local maxima: under all of the models, the genotype with all genes mutated is always reached with probability 1 as time goes to infinity (this is also the case for MHN, even if it can include inhibiting effects). To try to overcome this limitation without making specific additional assumptions about the sampling process, and for MHN, CBN, and MCCBN, we have used the uniformization method [52, 53] (see also Figure 6 of [3]) to approximate the continuous-time Markov chain ($\mathbf{Q}$) by a discrete time Markov chain (with transition matrix $= \mathbf{I} + \frac{1}{\gamma}\mathbf{Q}$, where $\gamma = \max(|q_{ii}|)$). We will interpret the diagonal entries of the time-discretized transition matrix as a lower bound on the

probability that the observed genotypes behave as local maxima, i.e. that no genotype conforms to our criteria of consecutiveness with respect to the observed one and therefore there is a non-zero probability to stay in it. The output from CBN, MHN, and MCCBN after time discretization will be referred to as CBN_td, MHN_td, and MCCBN_td, respectively, and we will differentiate this **time-discretized** results from those obtained, as explained above, using competing exponentials. We will refer to these methods as the TD methods (for time-discretized) whereas their counterparts, CBN, MHN, MCCBN, obtained via competing exponentials on the rate matrix, will be referred below as the CE methods. Notice that the transition probabilities given by a TD method are proportional to the ones given by its CE counterpart, except for the probability of staying in the current observed genotype which will necessarily be 0 in the CE variants. This difference, however, is critical for some fitness landscapes where fixation can occur at a fitness maximum (either local or global) that does not correspond to the genotype with all loci mutated. Importantly, the probablities that we are interpreting as corresponding to "staying in the observed genotype" are not arbitrary, but based on the underlying statistical models of the time-discretized methods. These may differ significantly in their predictions for staying in the observed genotype, which can give rise to differences in performance. OT, CAPRESE, and CAPRI do not belong to any of those two groups, though none of them can accommodate local maxima either; thus, when we refer to the "non-TD" methods, that includes MHN, CBN, MCCBN (in their "non-TD" versions) as well as OT, CAPRESE, and CAPRI.

In summary, a total of 13 methods have been used: CBN, in three variants (CBN, with transition probabilities using competing exponentials conditional on transition; CBN_uw, obtained from the former by using equiprobable transitions; CBN_td, the time-discretized version of CBN); MCCBN, again in three variants (MCCBN; MCCBN_uw; MCCBN_td); MHN in two variants (MHN and MHN_td); OT in variants OT and OT_uw (equiprobable transitions); CAPRESE; CAPRI, in two variants, CAPRI_AIC, CAPRI_BIC (which differ on the penalty used, AIC or BIC). Each of these methods returns a matrix $\hat{\mathbf{P}}$ of predicted transition probabilities.

## Quantification of similarity between predicted and true transition probabilities

For every observed genotype $i$ we have compared the vector of true probabilities $\mathbf{p}_i$, from the $i$th row of $\mathbf{P}$ with the analogous vector of predictions from a CPM $\hat{\mathbf{p}}_i$ returned from the $i$th row of $\hat{\mathbf{P}}$ for each CPM. We measured the similarity between these two vectors using the Jensen-Shannon distance ($JS$) [54], the square root of the Jensen-Shannon divergence [55]. The Jensen-Shannon divergence is a symmetrized Kullback-Leibler divergence between two probability distributions that is 0 when the two distributions are identical and reaches its maximum value of 1 if the two distributions do not overlap (we used log of base 2). See further details in section 1.4 of S1 Appendix.

## Factors that affect predictive performance: Linear mixed-effects model trees

To examine the relevance of the different factors on the performance of methods we used linear mixed-effects model trees. These are an extension of recursive partitioning (or tree-based) methods, where observations are first split repeatedly according to the predictor variables, which play the role of partitioning variables, so that the dependent variable becomes more homogeneous within each node; the linear mixed-effects model allows for the addition

**Table 1. Variables potentially affecting predictive performance.** Classification and description of the partitioning or predictor variables used in the linear mixed-effects model trees; these are the variables considered as possible sources of variation in the performance of CPMs. The abbreviations correspond to the variable names shown in Fig 2.

| Category | Variable | Abbreviation | Description |
|---|---|---|---|
| Fitness landscape and evolutionary dynamics-specific variables | Strong Selection Weak Mutation regime | `SSWM` | Measured as the average frequency of the most frequent genotype. |
| | Gamma ($\gamma$) | `gamma` | A measure of the amount of epistasis in the fitness landscape defined as "the correlation of fitness effects of the same mutation in single-mutant neighbors" [58] or, equivalently, "the correlation in fitness effects between genotypes that only differ by one locus, averaged across the [fitness] landscape" [59]. (Note: this is not the same gamma as used in the time-discretization of transition rate matrices.) |
| | Reciprocal sign epistasis | `epistRSign` | Fraction of pairs of loci with reciprocal sign epistasis. |
| | Number of observed peaks in the fitness landscape | `numObservedPeaks` | A measure of the ruggedness of the fitness landscape. |
| Sampling variables | Sample size | `sample_size` | Number of observations given to the models (50, 200 or 4000). |
| | Detection regime | `detect` | How the tumor was sampled with respect to the size distribution (*uniform* sampling: samples taken with equal probability through the evolutionary process; *small/large* sampling: samples taken predominantly early/late in the process). |
| Genotype-specific variables (within fitness landscape) | Number of mutations | `nMut` | Number of mutations of the observed genotype. |
| | Fitness rank | `fitnessRank` | Rank of the observed genotype, where the global maximum of the landscape has rank 1. |
| | Probability of the observed genotype being a fitness maximum | `propLocalMax` | Proportion of times where the observed genotype is a (possibly local) fitness maximum. |
| Genotype (within fitness landscape) by sampling by replicate-specific variables | Observability | `observedProp` | Proportion of times where the observed genotype was present in the sample provided to the models. |
| | Difference between observability and true frequency | `diff_obs_prop` | The true frequency is computed over the 20000 samples taken from each evolutionary process. |

of random effects [56, 57]. In our models, the dependent variable is the minimal *JS* over all methods (for each genotype by replicate by id by detection regime by sample size); this constitutes, therefore, the best possible performance over the set of all methods considered. The possible partitioning variables include characteristics that are fitness landscape- and evolutionary dynamics-specific, sampling characteristics, genotype (within fitness landscape)-specific, and genotype (within fitness landscape) by sampling (sample size and detection regime) by replicate-specific variables. The variables considered are described and classified in Table 1. Random effects for the global model are ID and, nested within ID, the crossed random effects of genotype and replicate. As the meaning of the number of mutations (`nMut`) is different in models with 7 and 10 genes, we fitted separate models for simulations with 7 and 10 genes. Models were fitted using the R package "glmertree" [56]. The Bonferroni-corrected significance level for node-splitting was set at 0.01. We specified a minimal (weighted) size in leaves (or terminal nodes) of 1%; as the returned trees had over 50 leaves in all cases, to allow for interpretability we pruned the resulting tree by recursively merging (starting from the leaves) all children node with a fitted minimal *JS* > 0.0677 (which corresponds to the 20% best *JS* –see Table D in S1 Appendix). In addition, we merged all children with a fitted minimal *JS* that differed by less than 0.0677/4 (so as to collapse good performing nodes with minor differences). Further details about the fitting procedure are provided in section 1.8 of S1 Appendix.

## Cancer data sets

We have used a total of 25 data sets. Twenty two of them were used in [25]. These 22 data sets include six different cancer types (breast, glioblastoma, lung, ovarian, colorectal, and pancreatic cancer), use different types of features (nonsynonymous somatic mutations, copy number alterations, or both) analyzed in terms of pathways, functional modules, genes, gene events, or mutations (yielding from 3 to 192 different features), and have samples sizes from 27 to 594; the original sources are [60–70] and all have been used previously in CPM research [5, 9–11, 25, 71–73]; for further details about those 22 data sets see [25] (their file S5_Text). We have also used three CGH data sets previously used in [3, 4] that were originally obtained from the Progenetix database [74]. These are `Breast-CGH`, from 871 breast cancer patients with 10 CGH alterations, `Renal-CGH`, from 251 renal cell carcinoma with 12 CGH alterations, and `Colon-CGH`, from 570 colorectal cancers with 11 CGH alterations. These three data sets were obtained from [3]. Thus, these 25 cancer data sets constitute a wide, representative example of data to which researchers have applied or might want to apply CPMs. We have analyzed all the data sets with all the methods considered; because computing time increases nonlinearly with number of features for some methods, and in particular CBN (and, to a lesser extent, MHN), have difficulties with 16 or more features, for the data sets with more than 15 features, we have used the common procedure [75] of selecting the 15 most frequent features.

Because we lack a "standard of truth" in the biological data (i.e. we do not know the true paths of cancer progession, unlike in the simulations, and instead have only a limited number of observations) we aimed to identify specific genotypes from which consecutive tumor progression could potentially be predicted accurately. We studied the similarities across predictions from different methods (measured as the Jensen-Shannon distance of the probability distributions returned by each one) for every feature in the biological data sets. We then compared this similarities with the patterns seen in the simulated data when good performance was possible.

## Results

### Simulation results

Fig 2 shows the liner mixed-effects model tree for fitness landscapes with seven genes under the uniform detection regime and with the largest sample size (4000). In Fig A in S1 Appendix (section 2.4.1) we show the results for 10 genes. Figs B and C in S1 Appendix present the models fitted to all detection regimes and sample sizes. For clarity, these figures do not show methods CBN_uw, MCCBN_uw, OT_uw: for methods with both a weighted and an unweigthed version (CBN_uw, MCCBN_uw, OT_uw) using weights never lead to worse performance and sometimes improved it, as can be seen in Figs D to G in S1 Appendix (section 2.4.2). In these figures we can see the relevance of detection regime and sample size: the smallest *JS* were generally achieved under the uniform detection regime and with the largest sample size (4000) (see also Tables A to G in S1 Appendix. Moreover, there were interactions between detection regime and sample size on the one hand and genotype-specific and fitness landscape-specific variables on the other; these interactions can make it harder to understand the relevance of fitness landscape-specific and genotype-specific variables on the quality of predictions. Fig 2 and Fig A in S1 Appendix isolate the effects of fitness landscape and genotype-specific variables, which is one of the main objectives of this paper.

The minimal *JS* over all methods considered (i.e., the best possible performance) generally improves in the representable fitness landscapes compared to the local maxima and RMF fitness landscapes, as can be seen in node 4 of Fig 2 (see also Tables B and C in S1 Appendix).

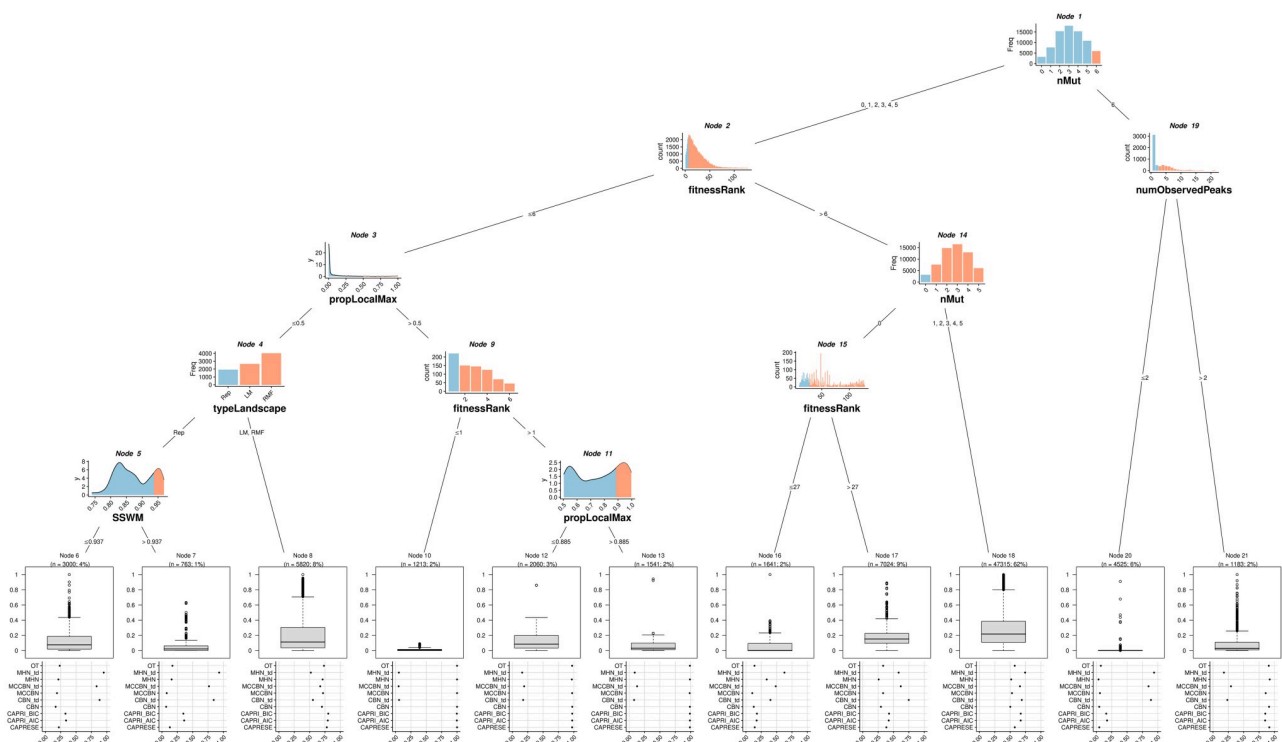

**Fig 2. Linear mixed-effects model tree fitted to data from fitness landscapes with 7 loci, sample size of 4000 and uniform sampling.** In the fitted models, the dependent variable is the minimal *JS* over all methods considered (i.e., the best possible performance): values closer to 0 represent better performance. Internal nodes show the partitioning variables and the values that lead to each partition; light blue denotes the values or categories that lead down the left split and salmon the values or categories that lead down the right split (partitioning criteria are also shown in the edges leading down a node). Leaves or terminal nodes show (weighted) boxplots of the dependent variable, the minimal *JS* over all methods considered, for the observations that fall in that leaf and, as label, the name of the node and the sample size (sum of weighted observations and % of the total data set for the fit). The dot plots underneath the boxplots show the (weighted) mean *JS* for each method for that leaf.

Moreover, gamma, reciprocal sign epistasis, and number of observed peaks, three of the variables that play important roles in the fitted trees (e.g., nodes 5, 11, 14, 15, and 23 in Fig B in S1 Appendix; node 5 in Fig A in S1 Appendix; nodes 12 and 15 in Fig C in S1 Appendix), perfectly separate the RMF from the rest of the fitness landscapes, and separate most local maxima fitness landscapes from the representable ones (Fig H in S1 Appendix). The results indicate, though, that highly specific fitness landscape characteristics (fitness landscape itself, gamma, reciprocal sign epistasis, number of observed peaks) show complex interactions with genotype-specific characteristics. For example, in Fig 2, fitness landscape plays a role in node 4 only for genotypes with small fitness rank (i.e., large relative fitness of a genotype compared to the rest of the genotypes in the fitness landscape) and that are rarely local maxima. Likewise, number of observed peaks in node 19, which would separate representable from local maxima and RMF landscapes, is relevant for the one-before-last genotype. Similar interactions between genotype-specific and fitness landscape-specific characteristics can be seen in the figures in the S1 Appendix (Fig B: node 23, child of genotype-specific nodes 1 and 19; Fig C: nodes 15 and 12, mediated by genotype-specific nodes 1, 11, and 14; Fig A: node 5, affected and which affects ancestor and descendant nodes of genotype-specific characteristics). A general observation is that deviations from SSWM lead to poorer predictive ability; these effects can depend both on general fitness landscape characteristics (e.g., node 5 in Fig 2) and genotype-specific characteristics (e.g., node 21 in Fig B in S1 Appendix and node 19 in Fig C in S1 Appendix).

Not all genotype-specific characteristics had a relevant effect. Observed proportion of a genotype and the difference between its observed proportion and true frequency in the sample are variables that do not appear in the final trees (although the former did appear in the models before prunning). Number of mutations had a non-linear effect: genotypes with either none or close to maximum number of mutations lead to better performance, whereas genotypes with intermediate number of mutations had worse performance (e.g., nodes 1 and 14 in Fig 2, nodes 1, 3, 6 in Fig A in S1 Appendix, nodes 2 and 6 in Fig C in S1 Appendix). This indicates that prediction becomes harder when the number of potential next genotypes increases, which happens at intermediate numbers of mutations. Smaller fitnessRank (i.e., large relative fitness of a genotype) was generally associated to better predictions (e.g., nodes 9 and 15 in Fig 2; nodes 12, 13, and 4 in Fig A in S1 Appendix) but, as with most other characteristics, the effect was strongly affected by other variables (node 1 in Fig B in S1 Appendix, nodes 11 and 21 in Fig C in S1 Appendix). Note that genotypes of fitness rank 1 are necessarily fitness maxima. The extremely poor performance of the non-TD methods in nodes 10, 12, and 13 in Fig 2 is in fact explained by the very large frequency of local maxima. Nodes 10, 12, and 13 correspond to genotypes with five or fewer mutations (all descend from the left split of node 1). The genotypes in node 10 are all local maxima (fitness rank of 1, with five or fewer loci mutated). Nodes 12 and 13 correspond to genotypes that are very often ($> 0.5$) or almost always ($> 0.885$) local maxima. For all cases in node 10, for virtually all cases in node 13, and for most of the cases in node 12, only the TD methods can do an acceptable job (predict local maxima) whereas the non-TD methods necessarily will predict a transition to another genotype, leading to *JS* values of 1.

In fact, how often a genotype is a local fitness maximum (propLocalMax) not only affected performance but also which methods were better. This is apparent in node 3 in Fig 2, node 2 in Fig A in S1 Appendix, node 19 in Fig B in S1 Appendix, and node 1 in Fig C in S1 Appendix: when a genotype was frequently (over 20 to 50%, depending on the data set) a local maximum, the TD methods lead to much better performance than the other methods. Moreover, once the frequency of local maxima was large, making it even larger lead to better performance (node 11 in Fig 2; nodes 14, 17, and 19 in Fig A in S1 Appendix; node 19 in Fig B in S1 Appendix; node 14 in Fig C in S1 Appendix). This phenomenon can also be looked at from the perspective of global fitness landscape characteristics: an increase in the number of local fitness peaks (numObservedPeaks) can lead to the TD methods being better than their CE counterparts (node 19 in Fig 2) or to the TD methods improving their performance (node 15 in Fig C in S1 Appendix). This is also observed with increasing reciprocal sign epistasis (episRSign—nodes 5, 11, 14 in Fig B in S1 Appendix), a parameter that can almost perfectly separate the RMF and local maxima landscapes from the representable ones (see Fig H in S1 Appendix). Note that epistRSign no longer plays a role in a switch between methods in node 23, since this node only splits genotypes in fitness landscapes that cannot be representable (genotypes that have a probability of being local maxima $> 0$ even when their fitness rank is $> 1$).

Scenarios with very good TD performance (*JS* less than approximately 0.1) comprise, then, between 1 and 5% of the cases (4% in nodes 10 and 14 in Fig 2; 5% in nodes 16, 20, and 21 in Fig A in S1 Appendix; 1% in node 24 of Fig B in S1 Appendix; 2% in nodes 18 and 22 of Fig C in S1 Appendix; see also Tables D to K of S1 Appendix). In these cases, MHN_td was generally slightly better performing than CBN_td (and MCCBN_td was generally the worse of the TD methods). Within the non-TD methods, when they could do well, most methods had roughly similar performance (except for CAPRI, that was normally the worse performer), with a slight advantage of CBN and MHN. CBN was generally among the best performers, and scenarios where MHN was the single best method were rare (but see node 20 in Fig 2). As with the TD methods, the CE methods were consistently very good only in limited scenarios (see also

Tables D to K in S1 Appendix. For CE methods these were nodes 7 and 20 in Fig 2 which comprised about 7% of the (weighted) observations, and slightly worse in nodes 6 and 16—where MHN's performance was surprisingly poor—that comprised another 6% of the observations. Similar patterns are seen in Fig A in S1 Appendix (node 23, with 5% of observations), Fig B in S1 Appendix (node 17, and with worse performance 6, 12, 16, comprising overall 6% of observations) and Fig C in S1 Appendix (node 10 and, with worse performance, 9, both with 1% of the observations).

### Target genotypes for consecutive prediction in real cancer data sets

What are the implications of the above results for the analysis of cancer data? We have compared the predictions from the best performing methods on 25 biological data sets: even if we do not know the true transition probabilities, we can examine the consequences of assuming that we choose one method when another method is the one that gives the best (or even perfect) predictions. This is what we do in Fig 3. The left panel of Fig 3A, for example, shows the difference in predictions (measured using *JS*) if the method that gave perfect predictions were CBN but we had used MHN (or viceversa). If we found that, for a given genotype, two or more methods gave similar predictions (i.e. comparing their predicted probabilities yielded a small *JS*) we could argue that they could all be potentially providing good results. On the other hand, very different predictions would indicate that at least one of them (if not all) is failing, and in that case there are no solid grounds to choose one or the other since we lack access to the underlying factors that affect performance. As can be seen, for most data sets there is wide variability in the consequences of using the wrong method, and often the difference in predictions is very large, specially if we use a TD vs. a non-TD method (MHN vs. MHN_td, right-most panel). The high variability between method predictions shown in Fig 3A—even across genotypes within a same dataset—highlights the importance of conditioning predictions to the observed genotype, as different methods can make very similar or substantially different

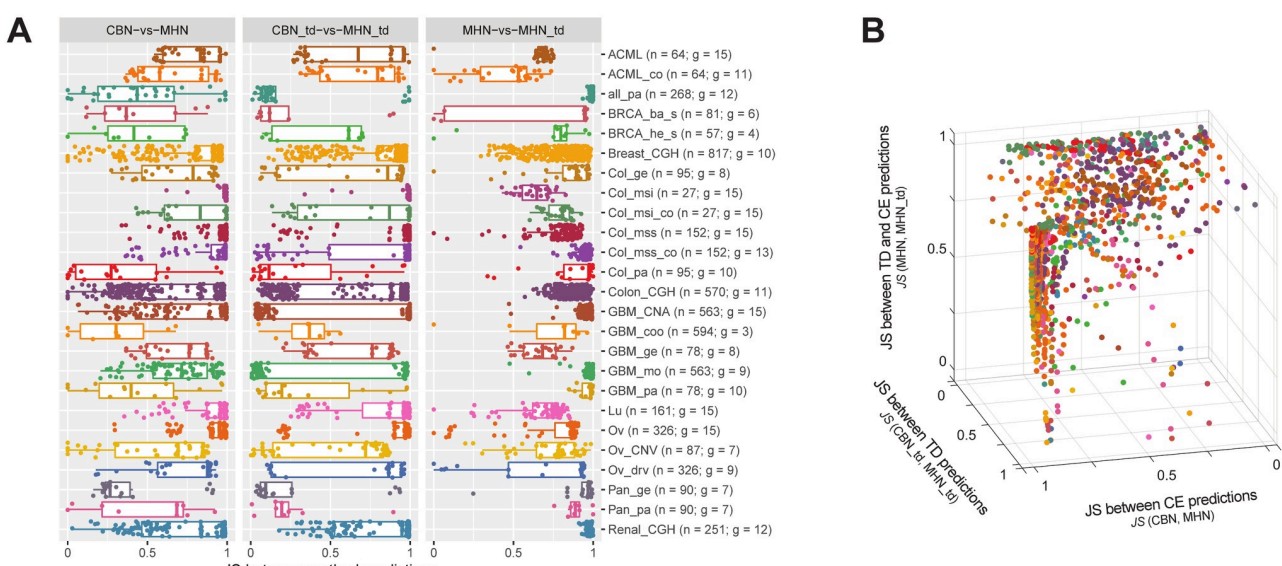

**Fig 3. Similarity between predictions of three pairs of methods for the 25 biological data sets.** For each data set, the *JS* divergence between the predictions of a pair of methods is represented in A: each of the three panels or B: each of the three axes. Dots correspond to all genotypes for which we computed predictions, which are the genotypes observed in the samples, except for the genotype with all genes mutated if it was observed. In parentheses, after the data set, we indicate the total sample size (n) and number of genes (g).

predictions depending on it. Particularly in the latter case, if any of the methods was indeed providing accurate predictions, determining which one is the correct one would depend on the properties of the observed genotype.

However, in general we do not have access to the underlying parameters that would allow us to make an informed decision on method (or method family) choice. So how do the above results contribute to the applied analysis of actual cancer data? Fig 3B shows the dissimilarity between the predictions of two specific CE methods (CBN and MHN), two specific TD methods (CBN_td and MHN_td), and two methods from different groups (MHN and MHN_td). Many genotypes accumulate in the area of the plot where the predictions of the two CE methods, as well as those of the two TD methods, are very different, with variable similarity between CE and TD predictions. But a few genotypes escape this trend. Looking back at the simulations, we have seen that there is a sharp difference in behavior of TD and CE methods when the best possible performance is good; and necessarily CBN and MHN (or CBN_td and MHN_td) can only both make correct predictions if both are making similar predictions. This same pattern—predictions of methods within the same group are similar and predictions of methods from different groups are not—is observed in the biological data for a small subset of genotypes (listed in section section 2.7 of S1 Appendix).

What about limiting our interpretation to those genotypes? This approach by itself will most likely yield only modest improvements in performance (see Tables H to K of S1 Appendix showing that, for TD, the proportion of cases with very good TD performance could at most increase from an unconditional 0.04 to a conditional 0.06, and the proportion of cases with very good performance of CE from an unconditional 0.04 to a conditional 0.16). Fig 4 shows the fraction of genotypes, for each data set, that meet fairly stringent conditions of similarity either within the CE methods, i.e. between CBN and MHN, or within the TD methods, i.e. between CBN_td and MHN_td (see figures in section 2.6 of S1 Appendix, for similarity for every individual genotype in each data set). For 11 of the data sets, there are more than 10% of the genotypes that fall in either of these categories (see Discussion).

## Discussion

Can we use cancer progression models (CPMs) to predict the short-term evolution of a tumor? Informally, we could word this question as "what genotype comes next in the path to tumor fixation?". More precisely, we can ask whether, conditional on us observing a particular genotype with $n$ mutations as the most abundant genotype in a tumor, we can predict what genotype with $n + 1$ mutations is in the Line of Descent (LOD) to the fitness maximum (where fixation occurs). We have examined the performance of 13 different CPM procedures for predicting short-term evolution, focusing mainly on the best possible performance over all 13 methods (i.e., on how close to the truth are the predictions of short-term evolution, regardless of method) and, secondarily, on the choice of method. The analysis of simulated data indicated that it was possible to provide predictions of short-term evolution that closely matched the true probabilities of short-term evolution only under some specific combinations of genotype characteristics and fitness landscape characteristics. And when good performance was possible, no method or set of methods was consistently superior, making choice of best method also dependent on genotype and fitness landscape properties.

Best possible performance was affected by complex interactions between fitness landscape-specific characteristics, deviations from the strong selection and weak mutation (SSWM) regime, sampling, and genotype-specific characteristics. The best performance was seen with large sample sizes, a uniform detection regime, with representable fitness landscapes and

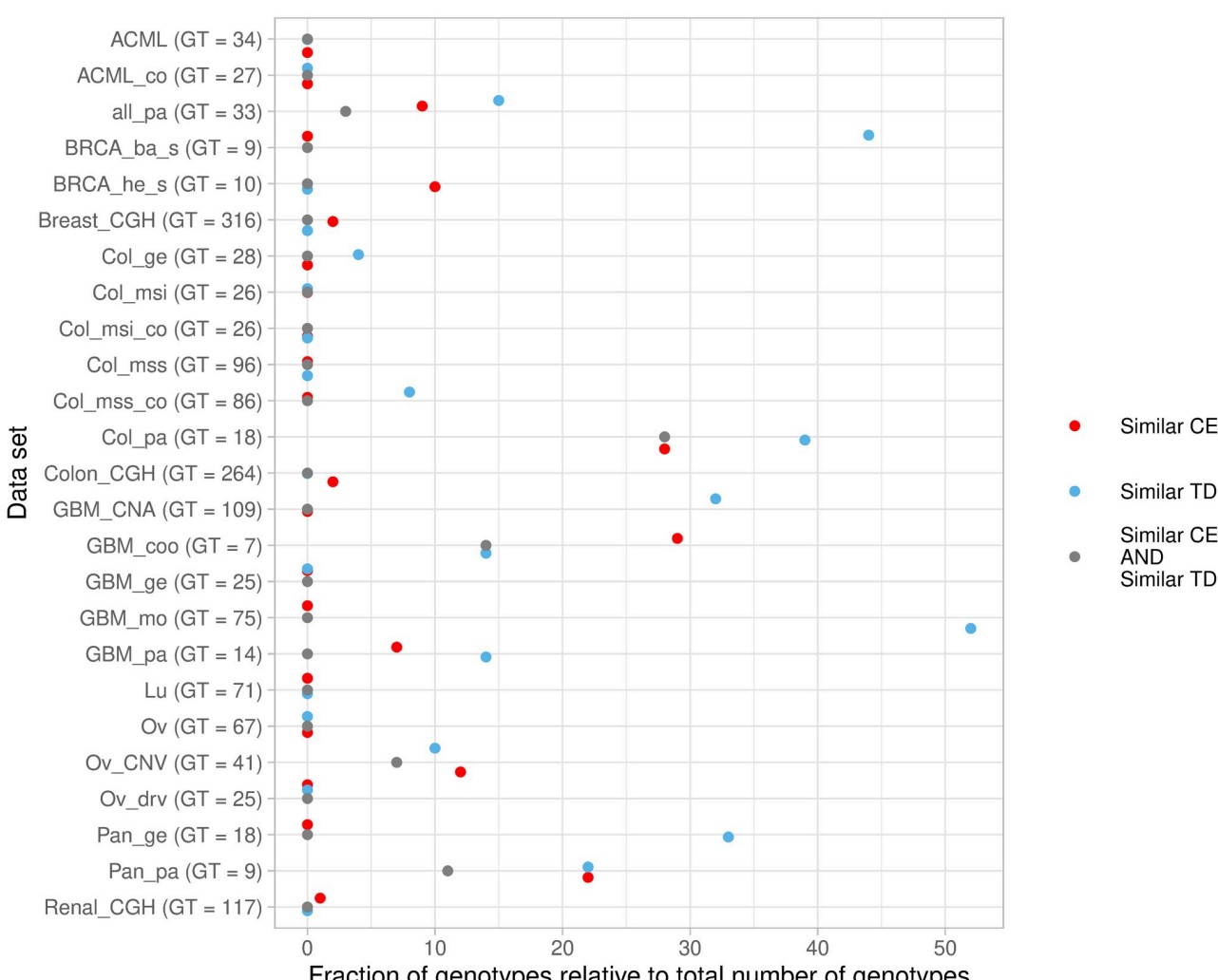

**Fig 4. Percentage of genotypes with similar TD and similar CE predictions for the 25 biological data sets.** For each data set, the red points show the percentage of genotypes in which predictions had both similar CE ($JS_{CBN,MHN} \leq 0.0677$) and different CE-TD ($(1/2)$ ($JS_{CBN\_td,CBN\_td} + JS_{MHN,MHN\_td}$) $\geq$ 0.7), the blue dots those with similar TD ($JS_{CBN\_td,MHN\_td} \leq 0.0677$) and different CE-TD, and the gray points the percentage of genotypes that had similar CE, similar TD, and different CE-TD. Values shown are the percentage relative to the total number of genotypes for which we computed predictions, which are the genotypes observed in the samples, except for the genotype with all genes mutated, if it was observed. In parentheses, after the data set, we indicate the total number of genotypes observed (GT).

under a SSWM regime. Genotypes with maximum or close to maximum fitness (i.e., small fitness rank) with either 0 or close to the maximum number of mutations, are those for which predictions tend to be better. For scenarios where the best possible performance was good (i.e., where it matters), these three factors also affected which methods were the best. There were sharply contrasting differences between methods, in particular, methods that can model local maxima—the TD methods MHN_td and CBN_td—tended to either both perform very well or both perform very bad. The corresponding two methods that do not model local maxima— MHN and CBN—generally present the opposite pattern, performing well when the TD methods do not and vice-versa.

Since our focus are predictions conditional on the observed genotype, we can also summarize the results by dividing genotypes into three groups: A) Genotypes for which using the TD methods lead to very good predictions ($JS < 0.1$); B) Genotypes for which the CE methods can

provide very good predictions ($JS < 0.1$); C) The remaining cases, which constitute the majority of the cases, where performance was generally poor ($JS > 0.25$). Group A consists of genotypes that are more often than not local maxima (probability of being a local maximum $> 0.5$) and of large fitness (small fitnessRank) or that are often local maxima (probability of being a local maximum $> 0.25$) while deviations from SSWM are minor. Fitness landscape-specific characteristics that make these genotypes more likely to be observed are large number of local fitness peaks and large reciprocal sign epistasis. For these genotypes, the TD methods can correctly predict being at the end of the evolutionary process, something that the remaining methods cannot account for, which explains the very poor performance of the non-TD methods in these cases ($JS > 0.75$). In group B are genotypes of large relative fitness, no mutations or with all except one loci mutated, in evolutionary scenarios with SSWM and smooth fitness landscapes (little reciprocal sign epistasis, large gamma) and few local peaks (less than 3). For these genotypes, the TD methods make very poor predictions ($JS > 0.75$) while MHN and CBN do particularly good jobs, but CAPRESE and OT, two methods that can only model simpler patterns of dependencies, generally perform well too. In the remaining cases (group C), no method can provide good predictions. Assessing whether we can achieve good performance and, if so, deciding which method to use, requires detailed information about the genotype, the fitness landscape, and the evolutionary dynamics. For some small subset of scenarios performance could be very good but it could also be dismal if we use the incorrect method.

Our results for short-term predictions highlight variables that have been found to be relevant for long-term prediction of the complete evolutionary paths. Sampling regime and sample size have previously been found relevant by [25, 26] (see also [51, 75] for the effect of sampling regime in inferring the DAGs of restrictions *per se*). Likewise, evolutionary scenarios with clonal interference in rugged fitness landscapes (small gamma, large reciprocal sign epistasis, large number of local fitness maxima) lead to poorer performance of CPMs compared to SSWM regimes in single-peaked fitness landscapes [25, 26]. The role of smooth fitness landscapes for CPM's predictive ability is directly related to the CPM assumption that final fixation occurs in the genotype with all loci mutated and their inability to model reciprocal sign epistasis [51] (this problem also affects MHN, specially if there are higher-order epistatic interactions). Interestingly, these are the fitness landscapes where the TD methods (which allow us to capture local maxima) perform better than the CE methods. The role of SSWM for CPMs has been discussed before by [25, 76] and [26] used it explicitly for predicting tumor evolution. In fact, all of the methods considered here implicitly assume SSWM mainly for two reasons. First, CPMs are generally applied to cross sectional data from bulk sequencing (not single-cell sequencing) without undergoing any deconvolution step (i.e. without attempting to infer the clonal composition of the sequenced tumor), which reflects the implicit assumption that an actual existing genotype can be obtained from bulk sequencing (see [77] for discussion of how this assumption can fail with bulk samples as we collapse over existing genotypes); if this assumption does not hold, the question "what genotype comes next" is undefined because we are conditioning on a non-existing genotype. Second, the observed genotypes are considered true steps in the path to tumor fixation, thus neglecting the possibility of tunneling effects [43–46]. These two assumptions are true if SSWM holds (see Fig 1, and note that the reverse is not necessarily true), in which case the most fit genotype at any given time is effectively the only one in the population, thus the only one observed, and an ancestor of the ultimately fixated one. Even when CPMs specifically include error estimations (e.g. [4]), these typically refer to sequencing error but not necessarily to deviations from SSWM. And our reformulation of the question "what genotype comes next" as we did in the Introduction allows us to deal with the second violation, but of course not with the first. In our analysis of the simulated data the first assumption holds (we sample true, existing genotypes); yet, our results show that using CPMs

to predict short-term evolution when the second assumption does not hold resulted in poorer performance. It must be emphasized, though, that the above comments refer to the effects of deviations from SSWM on the performance of the methods considered, not on predictability itself (the role of SSWM on predictability itself is more nuanced: see, e.g., [28, 30, 42–45, 78]).

In addition to the above factors, which affect both short- and long-term predictions, our results highlight that when we want to make conditional, short-term predictions, features of the genotypes for which we make the predictions can be as relevant as, or more relevant than, fitness landscape-specific features (see also [28–30] for a comparison of short-term vs. long-term predictability). We have already mentioned the combined importance of fitness rank and probability of being a local maximum, and the latter, together with reciprocal sign epistasis, illustrate factors that can increase short-term predictability while decreasing global-path predictability. The role of number of mutations, where predictions are worse from genotypes with intermediate numbers of mutations, can be understood as a consequence of the inverted U-shape relationship between the number of entries that can be non-zero in **P** and the number of mutations in a genotype ($\binom{N}{n+1}$: see section "Transition probabilities from evolutionary simulations" in the Methods): the problem of prediction becomes harder at intermediate numbers of mutations. The role of genotype-specific factors is a noteworthy feature because it suggests that CPMs might be leveraged to provide short-term predictions even when global, long-term predictions are difficult because fitness landscape- and evolutionary model-specific assumptions are violated.

As is common to simulation work, our results might be of limited value for the analysis of experimental data if our simulations fail to capture key aspects of tumor progression relevant for the problem addressed. Moreover, the analyses discussed above make use of full information from the simulated data that is of course not available in the analysis of real experimental data sets. In the absence of that information, for the analysis of the 25 cancer data sets, and to try to identify genotypes that show potential for accurate consecutive prediction, we used fairly stringent conditions of similarity of predictions within the CE methods (between CBN and MHN), or within the TD methods (between CBN_td and MHN_td). As seen in Fig 4, for 11 of the data sets, more than 10% of the genotypes meet these stringent conditions: which among the predictions that fulfill those similarity conditions should we trust? The difficulty of this choice is even more dramatic when the same genotype simultaneosuly shows large similarity within TD methods, large similarity within CE methods, and large dissimilarity between the CE and TD methods, as indicated by the gray points in Fig 4. In these cases, predictions between families of methods are very different and yet method (or method family) choice would require fitness landscape-specific, evolutionary model-specific and genotype-specific information. However, even if the information that would allow us to choose between predictions and methods is generally unknown in practice, our results provide a foundation to narrow down the search for genotypes for which good performance is possible. Furthermore, we have demonstrated that we do not need complete information on the general properties of the fitness landscape; instead, even partial information on those specific genotypes could serve to make informed decisions on method choice. Gathering information on the properties of individual genotypes is, in principle, a more reasonable objective than obtaining a full description of the complete fitness landscape for every cancer type. Our approach allowed us to identify specific genotypes in biological cancer data sets that show potential for accurate consecutive prediction. The two data sets with most genotypes exhibiting this potential (note that not all data sets contain any) correspond to colorectal cancer (Col_pa and Colon_CHG, see section 2.7 of S1 Appendix), perhaps indicating that tumor progression is particularly predictable (at least locally) in these. In all data sets, genotypes exhibiting this pattern generally have a high

mutational load (number of mutations between 0.6 and 0.9 times the total number of features in the data set), which could potentially mean they are more likely to behave as local fitness maxima –therefore the sharp differences in predictions from the TD and the CE methods, since only the former are able to model such local maxima.

Of course, validating these hypotheses would require a more careful analysis of local predictability in real cancer data sets. The lack of a standard of truth limits our ability to assess the performance of these methods in real clinical scenarios. One possible way to overcome this limitation would be to choose a proxy that can reflect the true transition probabilities. This could involve assesing the repeatability of specific steps in the paths of tumor progression from multi-region sequencing data [2, 21]. The performance of a CPM could be quantified as the overlap between the transition probabilities it predicts and the observed repeatability of the evolutionary steps corresponding to those transitions. These are possibly fruitful avenues for future research provided that data sets of large enough sample size for both cross-sectional and longitudinal analysis become available, or that the performance of CPMs is assesed when powered with multi-region sequencing data –where samples are not independent as multiple can correspond to a same evolutionary process, i.e. a same tumor (see section 2.7 of S1 Appendix for details).

## Conclusion

CPMs could, under very specific combinations of genotype characteristics and fitness landscape characteristics, be used to obtain good predictions of the short-term evolution of a tumor, even when long-term predictions are not possible because of violations of assumptions of CPM procedures. But method choice and assessing if the predictions obtained are to be trusted, for example to guide therapeutic decisions, requires detailed knowledge which is not available with empirical data. More generally, our work shows the promises and practical difficulties of using cross-sectional data in an evolutionary context to guide individualized therapies, even when cross-sectional data recapitulate the equivalent of many repeated evolutionary processes. Exploiting the potential of these methods will require further work to, first, examine the interpretation of their output and the consequences of their usage when key assumptions do not hold; second, identify if the methods' output, by itself or in combination with data set features, can provide indicators of performance; and, third, develop strategies to determine the characteristics of individual genotypes in real tumors and adapt method use consequently. For all three tasks, it will be important to make explicit the relationship between assumptions with respect to the evolutionary model (e.g., SSWM), fitness landscape characteristics (e.g., local maxima and reciprocal sign epistasis), sampling regimes (late vs. early tumors), and predictions, all in the context of well defined concepts such as lines of descent and for specified tasks such as interventions.

## Supporting information

**S1 Appendix. Supplementary methods and results.**
(PDF)

## Acknowledgments

We thank A. Zeileis for answers about the use of the glmertree package and R. Schill for answers about the use of MHN, and R. Guantes and C. Lazaro-Perea for comments on the manuscript.

## Author Contributions

**Conceptualization:** Ramon Diaz-Uriarte.

**Data curation:** Juan Diaz-Colunga, Ramon Diaz-Uriarte.

**Formal analysis:** Juan Diaz-Colunga, Ramon Diaz-Uriarte.

**Funding acquisition:** Ramon Diaz-Uriarte.

**Investigation:** Juan Diaz-Colunga, Ramon Diaz-Uriarte.

**Methodology:** Juan Diaz-Colunga, Ramon Diaz-Uriarte.

**Project administration:** Ramon Diaz-Uriarte.

**Resources:** Ramon Diaz-Uriarte.

**Software:** Juan Diaz-Colunga, Ramon Diaz-Uriarte.

**Supervision:** Ramon Diaz-Uriarte.

**Validation:** Juan Diaz-Colunga, Ramon Diaz-Uriarte.

**Visualization:** Juan Diaz-Colunga, Ramon Diaz-Uriarte.

**Writing – original draft:** Juan Diaz-Colunga, Ramon Diaz-Uriarte.

**Writing – review & editing:** Juan Diaz-Colunga, Ramon Diaz-Uriarte.

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
