## [Decision Letter · Decision Letter 0]

31 Aug 2021

Dear Dr. Diaz-Uriarte,

Thank you very much for submitting your manuscript "Conditional prediction of consecutive tumor evolution using cancer progression models: What genotype comes next?" for consideration at PLOS Computational Biology.

As with all papers reviewed by the journal, your manuscript was reviewed by members of the editorial board and by several independent reviewers. In light of the reviews (below this email), we would like to invite the resubmission of a significantly-revised version that takes into account the reviewers' comments.

We cannot make any decision about publication until we have seen the revised manuscript and your response to the reviewers' comments. Your revised manuscript is also likely to be sent to reviewers for further evaluation.

Sincerely,

Rainer Spang

Guest Editor

PLOS Computational Biology

Ville Mustonen

Deputy Editor

PLOS Computational Biology

Reviewer's Responses to Questions

**Comments to the Authors:**

Reviewer #1: SHORT SUMMARY

The paper analyses how well Cancer Progression Models (CPM) that are inferred from cross-sectional data can predict short-term progression for a given tumor genotype. To this end, the authors generate cross-sectional data from evolutionary simulations under varying conditions and provide these to five CPMs (CBN, MHN, OT, CAPRI, CAPRESE).

In particular, they allow for two important conditions that none of these CPMs can account for:

- violation of strong selection and weak mutation (SSWM) such that multiple clones can coexist at a time, whereas CPMs assume that clonal sweeps quickly take over the entire tumor.

- fitness landscapes with local maxima, whereas CPMs assume that the completely mutated genotype is always reached as time goes to infinity.

Under these conditions, the long-term predictions by CPMs for entire progression trajectories were already found to be poor in previous work. In this paper, the authors focus instead on short-term predictions for a given genotype, which is arguably more clinically relevant. They take great care in formulating this question rigorously as

"Given that a genotype with n mutations has been observed, what genotype with n+1 mutations is next in the path of tumor progression (Line of Descent)?”

which is well-defined whether SSWM does hold or not. This allows for an extensive comparison of predictive performance for different CPMs under varying simulation conditions.

GENERAL COMMENTS

The paper is generally well written and examines important questions for the tumor progression modelling community, bridging the gap between abstract cross-sectional models and more fine-grained evolutionary models. Although expanding upon similar previous work, the contribution is novel and relevant.

The analysis of predictive performance and how it depends on various conditions is extensive and well done. However, is not clear whether these simulations justify strong conclusions for real biological applications.

SPECIFIC COMMENTS, MAJOR:

1. Since CBN, MCCBN and MHN cannot account for fitness landscapes with local maxima, the authors additionally introduce time-discretized variants of these models from their uniformized transition rate matrices. It is not immediately clear to me why line 206 "we will interpret the diagonal entries of the time-discretized transition matrix as a lower bound on the probability that the given genotypes behave as local maxima" is justified biologically.

Moreover, it is not clear how the predictions from uniformization can differ from racing exponentials, other than allowing for a probability to stay in the current genotype. From supplementary section 1.1.1 I gather that staying in the current genotype is in fact a valid answer to "which genotype comes next?". If so, this should be made clear in the main manuscript.

I recommend expanding the paragraph in lines 201 - 215 with more detailed explanations.

2. The considered fitness landscapes seem somewhat limited to draw strong conclusions from. Rough Mount Fuji is essentially a single peak with additional random noise, and representable landscapes are those that correspond to CPMs with directed acyclic graphs (CBN and CAPRI) or trees (OT, CAPRESE). Since the analysis also includes MHN, which is a cyclic graph, I suggest investigating whether one can formulate a corresponding fitness landscape.

SPECIFIC COMMENTS, MINOR:

3. Section 2.6 "factors that affect predictive performance" is very difficult to read in prose. I suggest presenting this information as a table.

4. I don't see the motivation for unweighted versions of CPMs. Since line 348 "For methods with both a weighted and an unweighted version (CBN_uw, MCCBN_uw, OT_uw) using weights never lead to worse performance and sometimes improved it" indicates that they turned out irrelevant, I suggest dropping them from the analysis altogether to make it more assessable.

5. In line 117 "To examine the implications of our results for the analyses of cancer data, we have analyzed 25 cancer data sets, described in section Cancer data sets" I suggest adding a short description of the concrete analysis that was performed on the biological data sets.

Reviewer #2: In this manuscript, Diaz-Colugna and Diaz-Uriarte have examined the conditional predictability of short-term evolution of tumors using variety of Cancer Progression Models. Importantly, they have also quantified the effects of fitness-landscape and genotype-specific variables on the predictive performance using linear mixed-effects model trees. This study on its own is interesting and complements the author’s recent lines of research (Referenced in the manuscript as: [24], [25], [26] and [27]). However, the writing can substantially benefit from some major re-organizations, and I have some suggestions that might help further clarify the main points and so potentially enhance the overall quality of the paper.

1. My major concern is that the paper seems too technical, and its biological relevance is not well motivated. I strongly recommend the authors to elaborate on the biological significance of their work in the initial paragraphs of the introduction section. How does the predictability of the next genotype can lead to more precise therapeutic choice? Does higher predictability ensure more precise stratification of patients? Can they cite the relevant literature that have addressed these topics to some extent? One potentially inspiring paper that is relevant in this regard and the authors have not cited in the current manuscript is the concept of repeated evolutionary trajectories (Caravagna et al. 2018. Detecting repeated cancer evolution from multi-region tumor sequencing data. Nat. Methods. 2018, 15, 707-714.) in which the relevance of repeatability of cancer evolution in precision medicine is more clearly established.

2. The manuscript will certainly benefit from some major re-organizations, especially in the results and discussion sections. I recommend the authors to move their real data analysis from the discussion section (lines 445-489 including figures 3 and 4) to the results section and divide the results section into at least two subsections: 1. Simulation results, 2. Real data analysis. In this way, the authors will have better opportunity to provide forward-looking perspective on the biological relevance of their findings in the discussion section, which is currently missing to some extent.

3. I think the phrase “Best achievable performance” in line 297 (page 6, the last paragraph) is quite vague and the reader needs to be reminded what exactly it means. I suspect that it is defined in lines 236-237 at page 5: “In our models, the dependent variable is the minimal JS over all methods”. Similarly, I think including this definition in the caption of figure 2 will be beneficial, because I guess the JS values in the box plots are minimized across all methods.

4. In figure 2, especially in nodes 10 and 12, we can observe that the average JS score for most of the individual methods is almost one. Would the authors comment on how this is possible? To my view, this might either be a technical error or more likely an interesting observation that deserves to be further elaborated and discussed.

5. Obviously because of the lack of standard of truth, the authors have limited their real data analysis on comparing pairs of methods for calculating the JS scores. However, one way to overcome this limitation is to choose a proxy that can reflect the true transition probabilities. For example, the concept of repeated evolutionary trajectories that I mentioned above can be greatly helpful here (Caravagna et al. 2018. Detecting repeated cancer evolution from multi-region tumor sequencing data. Nat. Methods. 2018, 15, 707-714.). Are the evolutionary steps with higher transition probabilities overlap with the repeated evolutionary trajectories reported in the above study? Which of the 13 methods can retrieve a higher fraction of the repeated evolutionary trajectories?

Alternatively, the authors can further dig into the set of interesting genotypes that they have identified and described in lines 473-475: “This same pattern —predictions of methods within the same group are similar and predictions of methods from different groups are not— is observed in the biological data for a small subset of genotypes.”. Is it possible for the authors to catalogue those genotypes for each cancer type? Is it possible to detect some biologically interpretable patterns from those genotype sets? Can those genotypes be somehow connected to the concept of repeated evolutionary trajectories? Is there a chance to utilize them for precise stratification of patients and predict the outcome of therapies?

**Have the authors made all data and (if applicable) computational code underlying the findings in their manuscript fully available?**

Reviewer #1: Yes

Reviewer #2: Yes

PLOS authors have the option to publish the peer review history of their article (what does this mean?). If published, this will include your full peer review and any attached files.

Reviewer #1: No

Reviewer #2: **Yes: **Sayed Rzgar Hosseini
---

## [Decision Letter · Decision Letter 1]

25 Nov 2021

Dear Dr. Diaz-Uriarte,

We are pleased to inform you that your manuscript 'Conditional prediction of consecutive tumor evolution using cancer progression models: What genotype comes next?' has been provisionally accepted for publication in PLOS Computational Biology.

Best regards,

Rainer Spang

Guest Editor

PLOS Computational Biology

Ville Mustonen

Deputy Editor

PLOS Computational Biology

Congratulations, both reviewers are totally convinced by your revision of this nice work. Please address the minor comment of reviewer 2 concerning the section titles.

Rainer

The titles used for sections 2.7 and 3.2 are both the same ("Cancer data sets"). I suggest using a different and perhaps more informative title for section 3.2.

Reviewer's Responses to Questions

**Comments to the Authors:**

Reviewer #1: The authors have addressed all concerns to my satisfaction.

Reviewer #2: I thank the authors for adequately addressing my comments. Congratulations!

A Minor Comment:

The titles used for sections 2.7 and 3.2 are both the same ("Cancer data sets"). I suggest using a different and perhaps more informative title for section 3.2.

**Have the authors made all data and (if applicable) computational code underlying the findings in their manuscript fully available?**

Reviewer #1: Yes

Reviewer #2: Yes

PLOS authors have the option to publish the peer review history of their article (what does this mean?). If published, this will include your full peer review and any attached files.

Reviewer #1: No

Reviewer #2: **Yes: **Sayed-Rzgar Hosseini

---

## [Editor Report · Acceptance letter]

16 Dec 2021

PCOMPBIOL-D-21-00878R1 

Conditional prediction of consecutive tumor evolution using cancer progression models: What genotype comes next?

Dear Dr Diaz-Uriarte,

I am pleased to inform you that your manuscript has been formally accepted for publication in PLOS Computational Biology. Your manuscript is now with our production department and you will be notified of the publication date in due course.

With kind regards,

Olena Szabo
